# *TRA2B* Gene Splice Variant Linked to Seizures and Neurodevelopmental Delay: A Second Case Study

**DOI:** 10.3390/ijms242115572

**Published:** 2023-10-25

**Authors:** Olga Shatokhina, Valeriia Kovalskaia, Peter Sparber, Inna Sharkova, Irina Mishina, Vera Kuznetsova, Oxana Ryzhkova

**Affiliations:** 1Federal State Budgetary Institution “Research Centre For Medical Genetics”, 115478 Moscow, Russia; mironovich_333@mail.ru (O.S.); mikhailova.v.a@mail.ru (V.K.); psparber93@gmail.com (P.S.); sharkova-inna@rambler.ru (I.S.); akimova@med-gen.ru (I.M.); 2LLC “Evogen“, 115162 Moscow, Russia; vera.s@inbox.ru

**Keywords:** *TRA2B*, seizures, neurodevelopmental delay, novel mutation, mRNA analysis

## Abstract

In this study, we report a novel splice variant in the *TRA2B* gene identified in a patient presenting with seizures and neurodevelopmental delay. This paper represents the second investigation of pathogenic variants in the *TRA2B* gene in humans, reaffirming the conclusions of the initial study and underscoring the importance of this research. Comprehensive genetic testing, including whole genome sequencing, Sanger sequencing, and mRNA analysis, was performed on the proband and her parents. The proband harbored a de novo c.170+1G>A variant in the RS1 domain of Tra2β, which was confirmed to be pathogenic through mRNA analysis, resulting in exon 2 deletion and a frameshift (p.Glu13Valfs*2). The clinical presentation of the patient was consistent with phenotypes described in one of the previous studies. These findings contribute to the dissemination and reinforcement of prior discoveries in the context of *TRA2B*-related syndrome and highlight the need for further investigation into the functional consequences and underlying pathogenic mechanisms associated with *TRA2B* mutations.

## 1. Introduction

Neurodevelopmental disorders (NDDs) are a group of disorders that impact the development and functioning of the brain and exhibit extensive genetic and clinical variability [1]. The misregulation of alternative mRNA splicing is associated with various neurological disorders due to its vital involvement in neuronal development and mature neuronal function [2].

The Tra2β protein, also known as Transformer-2-beta, is a frequently studied RNA-binding protein involved in the regulation of alternative splicing [3]. Tra2β contains RNA-recognition motifs (RRMs) that enable it to bind to specific RNA sequences and two RS domains located at the N-terminus (RS1) and the C-terminus (RS2) of the protein. They facilitate protein–protein interactions and allow Tra2β to interact with other splicing factors and components of the spliceosome [4,5]. In a study using mice, it was demonstrated that Tra2β plays a crucial role in cortical neurogenesis and is essential for the appropriate development of the cortex. By utilizing mice with specific mutations in the *TRA2B* gene in the cortex, the researchers observed that the depletion of Tra2β leads to apoptosis of neural progenitor cells and disruption of the cortical plate [3].

The human *TRA2B* gene is located on chromosome 3q27.2 and generates various splice isoforms. The full-length transcript, NM_004593.3, contains nine exons and encodes the Tra2β-1 protein (NP_004584.1). Other isoforms are not well characterized [6]. One of these, *TRA2B* transcript variant 2 (NM_001243879.2), skips exon 2 and initiates a shorter open reading frame from a downstream ATG in exon 3. This truncated version, called Tra2β-3 (NP_001230808.1), retains the RRM and RS2 domains but lacks RS1. While Tra2β-1 is expressed broadly in the body, including the developing human brain, the Tra2β-3 isoform is more specific to certain tissues and undergoes developmental regulation [6,7].

Previously, a group of authors first associated germline variants in the *TRA2B* gene to a novel neurodevelopmental syndrome. This syndrome is characterized by severe intellectual disability or developmental delay, often accompanied by epilepsy, behavioral abnormalities, brain abnormalities, feeding disorders, growth retardation, visual abnormalities, and other less common features. In 11 different families, loss-of-function heterozygous variants in the *TRA2B* gene were found, mostly as de novo mutations. These variants were concentrated in the 5’ portion of the *TRA2B* gene and cause the disease through an apparently dominant negative mechanism. This mechanism involves the utilization of an alternative translation start site, resulting in the overexpression of a shorter Tra2β protein [6].

In this study, a novel de novo variant has been identified in the *TRA2B* gene in a patient with neurodevelopmental delay and seizures. We present the description of clinical features of the disease in this proband. Therefore, our study provides additional evidence reaffirming the role of pathogenic variants in *TRA2B* as a causative factor for the recently described seizures and neurodevelopmental delay syndrome.

## 2. Results

### 2.1. Clinical Evaluation

The proband (II-1), a 14-month-old female, was referred to the Research Centre for Medical Genetics because of seizures (Figure 1). Her parents were unaffected (Figure 2a). She was born from a first pregnancy by emergency cesarean section, due to weak labor and 8 h waterless interval, at a weight of 2560 g and length of 50 cm. Pregnancy was accompanied by a herpes infection, oligohydramnios, and pericardial effusion at 31 weeks, after which the fetus stopped gaining weight.

By the age of five months, the child’s development was approximately one month behind the typical schedule. At four months, she demonstrated the ability to hold her head up, smile, walk, and interact with objects using her hands. However, at five months old, the individual experienced pharmacoresistant epileptic seizures, specifically, infantile spasms, without any apparent triggering factors. These seizures led to the loss of acquired skills and the development of a head tilt to the right.

These seizures were initially controlled with the medication Synacthen Depot (tetracosactide) in combination with Topamax (topiramate) for four months. However, after discontinuing Topamax, the seizures recurred. Subsequently, the seizures were managed for five months with a ketogenic diet in combination with Ethosuximide, but they returned after this period. Currently, the child experiences seizures once a day, primarily at night.

The proband can be described as displaying severe developmental delay with global hypotonia. Motor milestones were significantly delayed, with no independent walking achieved. However, she did not display any behavioral issues. On a ketogenic diet, she has shown cognitive development progress. She now understands spoken speech and can associate it with actions. She is capable of sitting, standing, picking up toys, vocalizing, and engaging in various activities.

Electroencephalogram (EEG) results indicated hypsarrhythmia. Brain MRI of the proband at the age of five months, in a T2-weighted sequence, revealed abnormal paraventricular and subcortical white matter signals and an accumulation of cerebrospinal fluid over the frontotemporal areas. The volume and shape of the cerebral ventricles were within the normal range (Figure 1a). Cardiac malformations included patent foramen ovale and a mild stenosis of the left pulmonary artery. No feeding difficulties or sleep disturbances were reported, while information about digestive and vision problems was not provided. The individual had some skeletal anomalies, including cutaneous syndactyly (fusion) of the second and third toes and clinodactyly of the fifth finger. Additionally, the patient had nonspecific facial anomalies such as a high forehead, low-set ears with protruding lower edges, macrostomia, and macroglossia. Alopecia areata was also observed (Figure 1b).

### 2.2. Whole Genome Sequencing

Whole genome sequencing was carried out for the proband and proband’s parents. Comprehensive analysis was performed based on associating with neurodevelopmental diseases.

The only expected deleterious variant was a novel heterozygous c.170+1G>A (NM_004593.3) splice variant in the intron 2 of the *TRA2B* gene. This variant was absent in the proband’s parents, and it was not found in the Genome Aggregation Database (gnomAD v.2.1.1) nor among the samples of 2400 Russian patients’ exomes. Sanger sequencing confirmed a heterozygous state of the variant in the proband and its absence in both parents (Figure 2b). Paternity and maternity were also confirmed. Consequently, we could conclude the de novo origin of the variant in this family. However, we have not examined the germ cells of the parents and could not rule out germinal mosaicism.

### 2.3. mRNA Analysis

In order to prove the pathogenic role of the identified variant, we performed mRNA analysis. Amplicon separation by electrophoresis on a polyacrylamide gel and Sanger sequencing confirmed the deletion of exon 2 (134 bp) in one proband’s allele as a result of the c.170+1G>A variant and demonstrated no canonical transcript modification in the nonaffected parents (Figure 2c,d). The c.170+1G>A variant is considered to be deleterious, as it is predicted to lead to a frameshift with the translation of the truncated protein (p.Glu13Valfs*2).

As a result, the variant c.170+1G>A was classified as pathogenic according to the guidelines for massive parallel sequencing (MPS) data interpretation (criteria PVS1, PS2, PM2).

## 3. Discussion

Mutations in the *TRA2B* gene have previously been associated with the novel autosomal dominant neurodevelopmental syndrome. However, only one study has documented this association in 11 families with pathogenic or likely pathogenic *TRA2B* variants. In our investigation, we report a new de novo *TRA2B* mutation in a patient presenting with neurodevelopmental delay and seizures. This study not only strengthens the connection between *TRA2B* mutations and NDDs but also presents the description of the clinical features of the patient.

Tra2β protein is a frequently studied RNA-binding protein that plays a crucial role in the regulation of alternative splicing. In a study by Smith et al., all variants clustered in the 5′ part of Tra2β, upstream of an alternative translation start site responsible for the expression of the noncanonical Tra2β-3 isoform. RNA sequencing and Western blot analyses showed that these variants decreased the expression of the canonical Tra2β-1 isoform, whereas they increased the expression of the Tra2β-3 isoform, which is shorter and lacks the N-terminal RS1 domain. It has been suggested that loss-of-function variants clustered in the 5′ part of Tra2β cause a novel neurodevelopmental syndrome through an apparently dominant-negative disease mechanism involving the use of an alternative translation initiation site and overexpression of a shorter repressive Tra2β protein [6]. The heterozygous de novo variant c.170+1G>A, identified in intron 2 of the *TRA2B* gene of our proband, is also localized within the 5’ region of the protein (in the RS1 domain). mRNA analysis from primary fibroblast cultures demonstrated the deletion of exon 2 of the Tra2β-1 protein isoform that is predicted to cause a frameshift, leading to the translation of a truncated protein (p.Glu13Valfs*2). Three variants from the previous study (c.170+1del, c.170+2T>C, and c.171-2A>G) probably also disrupt the splicing process at the exon 2/3 boundary, and the effect of one of these variants on Tra2β protein was assessed using Western blot analysis (c.170+1del). Due to the fact that the variant we identified is localized in the same splice region of the gene, we hypothesize that the variant c.170+1G>A also causes a novel neurodevelopmental syndrome through an apparently dominant-negative disease mechanism involving the use of an alternative translation initiation site and overexpression of the shorter repressor protein Tra2β.

The clinical presentation of our patient aligns with the phenotypic characteristics described in previous study. The main features included severe developmental delay, global hypotonia, pharmacoresistant infantile spasms, nonspecific brain lesions observed via MRI, cardiac malformations, skeletal anomalies, and nonspecific facial anomalies. Remarkably, our patient exhibited an absence of behavioral abnormalities. This could potentially be attributed to the young age of the patient, suggesting that the developmental stage may have played a role in mitigating the emergence of such abnormalities. Further investigations are warranted to explore the intricate interplay between age-related factors and the phenotypic expression of this condition. We also identified additional clinical features not mentioned in the previous study: clinodactyly of the fifth finger and alopecia areata. The presence of clinodactyly suggests that it may be a component of this syndrome, as hand skeletal anomalies have been reported before. We did not find any associations between the *TRA2B* gene and hair loss in the available literature. One factor that we consider as potentially linked to alopecia is the use of the antiepileptic drug Topamax. It is noteworthy that hair loss is a documented side effect of Topamax and may manifest in some patients using this medication [8]. However, further research is necessary to conduct a more comprehensive investigation into the relationship between Topamax usage and alopecia in the context of this syndrome. Additional studies, involving a larger patient sample and genetic data analysis, as well as consideration of other potential factors, can assist us in more precisely determining the potential causes of alopecia and its potential connection to the *TRA2B* gene.

In this case, it is important to highlight the significance of performing whole exome/genome sequencing on the trio, which includes the proband and her parents. This approach not only enhances the diagnostic yield but also facilitates the identification of variants in previously undescribed genes, including those for which there are single studies and are not yet represented in widely used databases such as HGMD and OMIM. This underscores the necessity of considering a broader genetic landscape beyond conventional gene sequencing panels, particularly in cases where the causal gene is not readily apparent or when studying neurodevelopmental disorders.

## 4. Materials and Methods

### 4.1. Clinical Data

The proband was examined at the Research and Counseling Department of the Research Centre for Medical Genetics (RCMG). The patient’s parents provided oral and written consent for this study, approved by the Ethics Committee of RCMG (approval number 4/1 from 19 April 2021). Her initial assessment included a full history and physical examination. MRI and EEG were also performed. 

### 4.2. Genome Sequencing

Blood samples from the proband and unaffected parents were collected, and genomic DNA was extracted using standard methods. Whole genome sequencing was performed for the proband and her parents. Samples were prepared using MGIEasy stLFR Library Prep Kit following the manufacturer’s protocol (MGI Tech Co., Ltd., Shenzhen, China). Sequencing was carried out using paired-end reading (2 × 100 bp) on the DNBSEQ-T7 (MGI Tech Co., Ltd., Shenzhen, China). Sequencing data were processed using NGSData software (ngs-data-ccu.epigenetic.ru, Moscow, Russia) (accessed on 24 October 2023). Average coverage for these samples is more than ×34. The sequenced fragments were visualized using Integrative Genomics Viewer (IGV) software v.2.16.2 (© 2013–2018 Broad Institute and the Regents of the University of California, San Diego, CA, USA). Filtering of the variants was based on their frequency—less than 1% in gnomAD v.2.1.1 —and coding region sequence effect: missense, nonsense, coding indels, and splice sites. The variants’ clinical significance was evaluated according to the guidelines for MPS data interpretation [9].

Amplification and Sanger sequencing were performed to validate the genome variant in *TRA2B* gene in the proband and her parents. To amplify the fragment encompassing the candidate variants, custom primers were used (according to the NM_004593.3 reference sequence): *TRA2B*_2F: ACCCTCTCTACCTTCCTGGTTT; *TRA2B*_2R: GGATTGCTGTCCCTCACCATTT.

### 4.3. RNA Analysis

Primary fibroblast cultures were established out of skin biopsies of the affected proband and unaffected parents in accordance with standardized operating procedure. The cell cultures were cultured in DMEM growth medium supplemented with 15% fetal bovine serum at 37 °C in an incubator with 5% CO2. Upon reaching the appropriate confluence, the cells were trypsinezed, washed with PBS, and used for subsequent total RNA extraction with QIAzol Lysis Reagent^®^ (Qiagen, Venlo, The Netherlands). The quality of extracted total RNA was evaluated by means of electrophoresis in agarose gel and spectrophotometric estimation of the 260/280 ratio during the total RNA concentration measurement. The reverse transcription of the total RNA samples was performed using the QuantiTect Reverse Transcription Kit^®^ (Qiagen, Venlo, The Netherlands). Standard PCR for cDNA samples of the affected patient and nonaffected parents was carried out using the following primers at 5‘UTR and exon 7 of the *TRA2B* gene (NM_004593.3): *TRA2B*_5‘F: AGAGGTTGGCAGCTTCGATT; *TRA2B*_7R: CTGATCCCTGTCTTGGGCAG.

The produced amplicons were identified through their separation by electrophoresis on a polyacrylamide gel and were subjected to Sanger sequencing for further analysis.

## 5. Conclusions

We report a novel case of *TRA2B*-associated neurodevelopmental disease, contributing to the dissemination and reinforcement of prior findings, and further enhancing the comprehension of this recently described pathology.

## Figures and Tables

**Figure 1 ijms-24-15572-f001:**
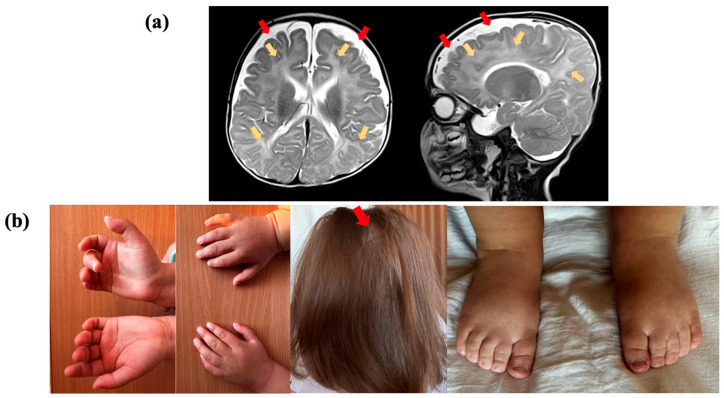
Clinical presentation of the proband, who had a variant in the *TRA2B* gene: (**a**) Brain magnetic resonance imaging (MRI) of the proband at the age of 5 months, in T2-weighted sequence, reveals abnormal paraventricular and subcortical white matter signals (yellow arrows) and the accumulation of cerebrospinal fluid over the frontotemporal areas (red arrows); (**b**) proband showing cutaneous syndactyly of the 2nd and 3rd toes, clinodactyly of the 5th finger, and alopecia areata.

**Figure 2 ijms-24-15572-f002:**
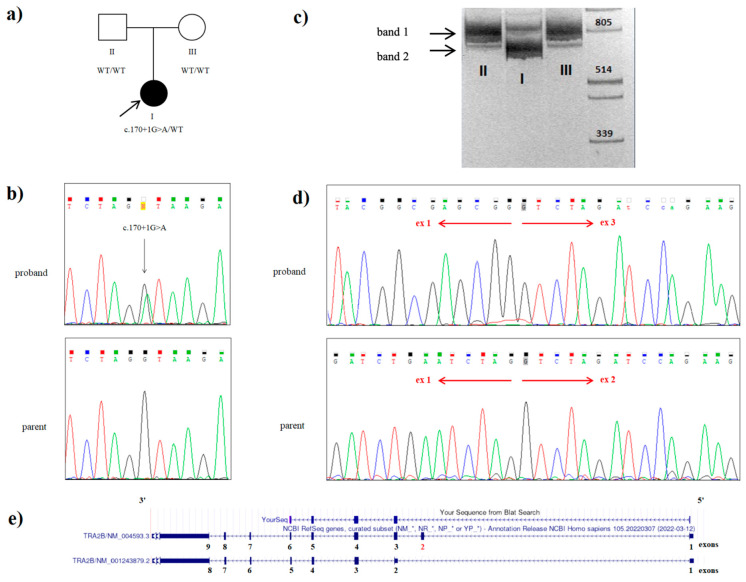
Genetic testing of the proband with a variant in the *TRA2B* gene: (**a**) the family tree showing the affected proband (I) and unaffected parents (II and III); (**b**) Sanger sequencing of the DNA results demonstrating the heterozygous c.170+1G>A splice *TRA2B* mutation in the proband; (**c**) visualization of cDNA amplification products for the proband (I) and her parents (II and III) with primers flanking 5‘UTR and exon 9 of the *TRA2B* gene. Band I (828 bp) includes the sequences of exons 1, 2, 3, 4, 5, 6, 7, 8, and 9 of the *TRA2B* gene, which corresponds to the canonical protein-coding transcript 1. Band II (694 bp) includes the sequences of exons 1, 3, 4, 5, 6, 7, 8, and 9 of the *TRA2B* gene without exon 2. This amplicon was present in the proband (I) and absent in her parents (II and III); (**d**) Sanger sequencing of the proband’s band 2, extracted from the gel and father’s cDNA amplicon: the electropherograms shows the presence of the aberrant *TRA2B* transcript with the exon 2 skipped in patient, while his father possesses only the canonical *TRA2B* transcript variant; (**e**) alignment of the patient’s cDNA sequence in the UCSC genomic browser.

## Data Availability

Reported variant can be found in ClinVar (Submission number: SCV004032532).

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
