# Peer review of "TRA2B Gene Splice Variant Linked to Seizures and Neurodevelopmental Delay: A Second Case Study"

_ijms, 2023, doi:10.3390/ijms242115572_

Round 1

Reviewer 1 Report

In this study, we report a novel splice variant in the TRA2B gene identified in a patient

presenting seizures and delay in neurological development. This article represents the second investigation

  of pathogenic variants in the TRA2B gene in humans, reaffirming the conclusions of the initial studies and underlining the importance of this research.

The article is very simple, and a continuation of a previous one.

The authors are modest in their conclusions, pointing out: We report a novel case of TRA2B-associated neurodevelopmental disease, adding to the emerging understanding of this recently described pathology. Our findings contribute to the expanding knowledge in this field and lay the foundation for further investigations.

In my opinion it can be published to disseminate and strengthen previous results, although I believe that a case study should appear in the title.

Author Response

Dear reviewer! Thank you for your feedback.

We have taken your comments into consideration and have made revisions to both the Abstract, Introduction and Conclusions sections (Lines 19-22, 61-63, 254-256). We have emphasized the importance of our work in contributing to the dissemination and reinforcement of prior findings, as evident in our conclusions. We have also updated the title of the article to better reflect the significance of our research in the context of the existing body of knowledge: «TRA2B Gene Splice Variant linked to Seizures and Neurodevelopmental Delay: A Second Case Study.» 

We appreciate your insights and hope that these adjustments enhance the presentation of our results.

All changes are highlighted in yellow. The corrections requested by another reviewer are highlighted in green and blue.

Reviewer 2 Report

In the manuscript " IDENTIFICATION OF A NOVEL SPLICE VARIANT IN THE TRA2B GENE ASSOCIATED WITH SEIZURES AND NEURODEVELOPMENTAL DELAY: A SECOND INVESTIGATION” by Drs Olga Shatokhina et al authors presented impressive    study of pathogenic variants of the TRA2B gene. I have no objections to the essence of the study; the data obtained are beyond doubt. There are just some clarifying questions.

Thus, the connection between alopecia and seizures has been studied previously. The authors write that the connection between them remains unclear, but the authors’ opinion would be interesting to readers. Perhaps it would be appropriate to add a few words regarding the possible mechanisms of hair growth regulation, the gene studied and epilepsy.

Regarding the data from the MRI studies, they look very brief. Are the volume and shape of the cerebral ventricles normal, or are there pathological changes? If yes, which ones? Is there any data on the ratio of white and gray matter?

The   manuscript is organized well, and written clearly. I will be happy to recommend the manuscript for the publication after minor correction, suggested before.

Author Response

Dear reviewer! Thank you very much for the thorough analysis of our article. I think all the corrections will benefit the article. All changes are highlighted in green. The corrections requested by another reviewer are highlighted in yellow and blue. Corrections to similar comments are highlighted in blue.

  • Thus, the connection between alopecia and seizures has been studied previously. The authors write that the connection between them remains unclear, but the authors’ opinion would be interesting to readers. Perhaps it would be appropriate to add a few words regarding the possible mechanisms of hair growth regulation, the gene studied and epilepsy.

Answer: Thank you for your insightful feedback. While our initial literature search did not reveal any associations between alopecia and seizures, we further analyzed the patient's treatment history and found that the medication Topamax could potentially induce alopecia, as it is a documented side effect. Given this information, we have refrained from referring to it as an additional clinical feature (Lines 193-202). This insight suggests that Topamax may be a more likely contributor to alopecia in our study, as opposed to the TRA2B gene. We appreciate your attention to this aspect of our research.

  • Regarding the data from the MRI studies, they look very brief. Are the volume and shape of the cerebral ventricles normal, or are there pathological changes? If yes, which ones? Is there any data on the ratio of white and gray matter?

Answer: The MRI data section has been expanded to provide a more comprehensive overview of the findings (please see Section Results (Lines 90-94) and Figure 1 (Lines 114-117)). The volume and shape of the cerebral ventricles were indeed within the normal range.

  • The   manuscript is organized well, and written clearly. I will be happy to recommend the manuscript for the publication after minor correction, suggested before.

Answer: Thanks a lot! We are very pleased that you highly appreciated our work.

Reviewer 3 Report

The authors report a novel splice site variant in TRA2B gene identified by genome sequencing in a patient with a neurodevelopmental delay and seizures.

The clinical features of the patient herein described, overlap the picture reported in the previous   report  with the first germline  mutations in TRA2B,  with few additional signs such as  clinodactyly of the 5th finger and alopecia.

Unlike previously reported patients,  no behavioral abnormalities  were observed in the patient herein described.

It would be interesting if the Authors provided some more details on the neuro-psychiatric aspects and the course of the patient's epileptic encephalopathy between 5 and 14 months.

The methodological  approach is appropriate,  however  few additional experiments would improve the quality of the data presented and would allow a more precise assessment of the patient's genetic condition.

The splicing variant c.170+1G>A  here reported affects  nucleotides already involved in other mutations, c.170+1del, c.170+2T>C and c.171-2A>G.  This supports the pathogenicity of the novel variant identified.  The authors observe that the variant is not present in any of the parents  and conclude that the variant is probably de novo.

Did  the authors carried out  paternity test?

The authors state that they cannot exclude mosaicism, however an exome sequencing with a good quality coverage allows to exclude mosaicism events. Maybe the authors could try to address this relevant issue.

Author Response

Dear reviewer! Thank you very much for the thorough analysis of our article. I think all the corrections will benefit the article. All changes are highlighted in blue. The corrections requested by another reviewer are highlighted in yellow and green.

  • It would be interesting if the Authors provided some more details on the neuro-psychiatric aspects and the course of the patient's epileptic encephalopathy between 5 and 14 months.

The methodological  approach is appropriate,  however  few additional experiments would improve the quality of the data presented and would allow a more precise assessment of the patient's genetic condition.

Answer: Thank you for your valuable suggestions. We have incorporated additional details on the neuro-psychiatric aspects and the course of the patient's epileptic encephalopathy in the Results Section. Furthermore, we have included supplementary brain MRI data in Section Results, which is also presented in Figure 1 (Lines 72-117). We hope that these enhancements improved the overall quality of the data presented.

  • The splicing variant c.170+1G>A here reported affects  nucleotides already involved in other mutations, c.170+1del, c.170+2T>C and c.171-2A>G.  This supports the pathogenicity of the novel variant identified.  The authors observe that the variant is not present in any of the parents  and conclude that the variant is probably de novo. Did  the authors carried out  paternity test?

Answer: Yes, we conducted paternity and maternity tests, and this information has been included in the text (Line 127). We appreciate your attention to this aspect of our study.

  • The authors state that they cannot exclude mosaicism, however an exome sequencing with a good quality coverage allows to exclude mosaicism events. Maybe the authors could try to address this relevant issue.

Answer: We would like to clarify that while our study did not detect mosaicism in the sequencing data, it is essential to note that we analyzed blood cells. Genome sequencing with high-quality coverage is indeed a powerful tool for detecting mosaicism events in a given tissue. However, germine mosaicism may still exist in other tissues or cell types not covered by our analysis.